# Review of Single Crystal Synthesis of 11 Iron-Based Superconductors

**DOI:** 10.3390/ma16144895

**Published:** 2023-07-08

**Authors:** Qiang Hou, Longfei Sun, Yue Sun, Zhixiang Shi

**Affiliations:** School of Physics, Southeast University, Nanjing 211189, China; 230208711@seu.edu.cn (Q.H.); 220222225@seu.edu.cn (L.S.)

**Keywords:** single crystal, hydrothermal, CVT, self-flux, annealing, 11 iron-based superconductors, superconductivity

## Abstract

The 11 system in the iron-based superconducting family has become one of the most extensively studied materials in the research of high-temperature superconductivity, due to their simple structure and rich physical properties. Many exotic properties, such as multiband electronic structure, electronic nematicity, topology and antiferromagnetic order, provide strong support for the theory of high-temperature superconductivity, and have been at the forefront of condensed matter physics in the past decade. One noteworthy aspect is that a high upper critical magnetic field, large critical current density and lower toxicity give the 11 system good application prospects. However, the research on 11 iron-based superconductors faces numerous obstacles, mainly stemming from the challenges associated with producing high-quality single crystals. Since the discovery of FeSe superconductivity in 2008, researchers have made significant progress in crystal growth, overcoming the hurdles that initially impeded their studies. Consequently, they have successfully established the complete phase diagrams of 11 iron-based superconductors, including FeSe_1−*x*_Te*_x_*, FeSe_1−*x*_S*_x_* and FeTe_1−*x*_S*_x_*. In this paper, we aim to provide a comprehensive summary of the preparation methods employed for 11 iron-based single crystals over the past decade. Specifically, we will focus on hydrothermal, chemical vapor transport (CVT), self-flux and annealing methods. Additionally, we will discuss the quality, size, and superconductivity properties exhibited by single crystals obtained through different preparation methods. By exploring these aspects, we can gain a better understanding of the advantages and limitations associated with each technique. High-quality single crystals serve as invaluable tools for advancing both the theoretical understanding and practical utilization of high-temperature superconductivity.

## 1. Introduction

The discovery of iron-based superconductivity represents a significant breakthrough in the field of condensed matter physics, with a profound impact on the study of high-temperature superconductivity [1,2]. According to the different types and ratios of elements in the parent compositions, it can be divided into several different types, such as 111, 122 and 1111 of the iron-pnictide superconductors and 11 and 122 of the iron-chalcogenide superconductors. These materials exhibit a wide range of fascinating physical phenomena, including a multi-band structure, an extremely small Fermi energy, and the presence of nematic and antiferromagnetic (AFM) ordered states. These unconventional superconducting properties make them prime candidates for exploring high-temperature superconductivity and its related properties [3,4,5]. Importantly, the unconventional superconductivity observed in iron-based materials cannot be explained by the conventional electron–phonon pairing mechanism. This breakthrough challenges the notion that cuprates are the sole class of high-temperature superconductors, thereby stimulating further research into the pairing mechanisms underlying high-temperature superconductivity [6,7,8].

Compared with FeAs-based superconductors, the 11 iron-based superconductors in iron-chalcogenide compounds have the advantages of a simple crystal structure and non-toxicity. FeSe consists solely of edge-sharing tetrahedral FeSe_4_ layers stacked along the *c*-axis, without a charge storage layer [9,10,11]. A structural transition from tetragonal to orthorhombic occurs at about *T*_s_ ~ 90 K accompanied by the nematic phase [12,13,14,15]. Despite having a relatively low superconducting critical temperature (*T*_c_) of approximately 9 K, high tunability and nematicity without magnetic order have garnered significant attention and research interest. Under high pressure, the *T*_c_ of FeSe can be elevated to approximately 38 K, and a new magnetic order emerges within a specific pressure range once the nematic phase is suppressed [16,17,18,19]. Chemical methods, such as intercalation [20,21], ionic liquid gating [22,23,24] and potassium deposition [25,26], have been employed to raise the *T*_c_ to over 40 K. Remarkably, monolayer FeSe films on doped SrTiO_3_ substrates have exhibited superconductivity with the *T*_c_ surpassing 65 K [27,28]. These materials offer various pathways to achieve a high *T*_c_ and exhibit unconventional superconducting behavior. Consequently, they have become pivotal in advancing research in the field of high-temperature superconductivity, playing a vital role similar to that of copper-based superconductors.

The substitution of isovalent sulfur (S) in FeSe, equivalent to applying positive chemical pressure, has proven to be an effective method for tuning superconductivity and nematic order. With S doping, the nematic transition temperature *T*_s_ gradually decreases until it vanishes at *x* ~ 0.17, marking a nonmagnetic nematic quantum critical point (QCP) [29,30,31,32]. Nuclear magnetic resonance (NMR) measurements indicate a strong suppression of AFM fluctuations with S substitution, resulting in negligible AFM fluctuations near the QCP [31]. Within the nematic regions, the *T*_c_ exhibits a small superconducting dome, reaching a maximum of 11 K at *x* ~ 0.11. Beyond the nematic regions, superconductivity is gradually suppressed, reaching a minimum at *x* ~ 0.45, after which the *T*_c_ slowly increases until *x* = 1 [33]. Notably, unlike when external pressure is applied, no new magnetic order emerges after the nematic phase [33,34,35]. 

Similarly, the substitution of isovalent tellurium (Te) in FeSe, equivalent to applying negative chemical pressure, is an effective method for tuning the superconductivity and various ordered states. In FeSe_1−*x*_Te*_x_* single crystals phase diagram, *T*_s_ linearly decreases until it disappears at *x* = 0.5 with Te doping [36,37,38]. The *T*_c_ initially decreases to a minimum at *x* ~ 0.3 and then increases to a maximum at *x* ~ 0.6; subsequently, the *T*_c_ is gradually suppressed and antiferromagnetic behavior emerges when *x* > 0.9 [36,37,39,40,41,42]. FeTe undergoes a tetragonal-to-monoclinic structural transition at around 70 K, exhibiting AFM behavior without superconductivity, reminiscent of the emergence of superconductivity from AFM in the cuprate superconductors [43,44,45,46,47]. The unique phase diagram of 11 iron-based superconductors, with its interplay of competing orders, nematic phase, magnetic order and superconductivity, provides important insights for exploring the mechanism of high-temperature superconductivity.

Unfortunately, preparing high-quality single crystals is one of the challenges in the study of the 11 iron-based superconducting system, particularly FeSe_1−*x*_Te*_x_* and FeSe_1−*x*_S*_x_*. This difficulty is also commonly encountered in the study of other iron-based superconducting families. On the one hand, the low chemical stability of FeSe_1−*x*_S*_x_* and the issue of phase separation in FeSe_1−*x*_Te*_x_* (0 < *x* < 0.5) make it arduous to obtain single crystals or single-phase samples using traditional solid-state reactions [48,49,50,51,52]. On the other hand, even though the preparation of single crystals of FeSe_1−*x*_Te*_x_* (0.5 ≤ *x* ≤ 1) is relatively straightforward using the self-flux method, the presence of excess Fe significantly affects the investigation of their intrinsic properties, such as the localization of charge carriers [53,54,55], spin glass phase [56] and incoherent electronic states [54,57]. It is difficult to prepare high-quality single crystals of the 11 system using traditional solid-state reaction methods, and new methods are gradually developed. 

To synthesize high quality single crystals across the entire doping range, different methods need to be employed. In this review, we provide an overview of the common synthesis methods for the 11 iron-based system, focusing on the optimal method for different doping regions, along with a relevant phase diagram of the entire region. Initially, we discuss the conventional methods of obtaining FeSe single crystals, namely the flux method and chemical vapor transport (CVT). In Section 3, we describe the preparation of FeSe_1−*x*_S*_x_* single crystals using CVT for range 0 ≤ *x* ≤ 0.29 and the hydrothermal method for the entire region. In Section 4, we explain how high quality FeSe_1−*x*_Te*_x_* (0 ≤ *x* ≤ 0.5) single crystals can be directly synthesized via CVT. For the Te-high doping region (0.5 < *x* ≤ 1), it becomes necessary to anneal the as-grown single crystals in O_2_ or Te vapor. Finally, we conclude the review with a summary and outlook in Section 6.

## 2. Single Crystal Growth and Superconductivity of FeSe

FeSe stands out as one of the most extensively studied materials within the realm of iron-based superconductors, owing to its array of unique properties. Notably, FeSe exhibits a multiband electronic structure, a nematic phase, a BCS-BEC crossover, and spin-density wave (SDW) behavior, all of which benefit from the use of high-quality single crystals. FeSe is considered a multiband compensated semimetal with a Fermi surface consisting of *d_xy_*, *d_yz_*, and *d_xz_* orbitals, forming well-separated electron and hole pockets [58]. Because the extremely small Fermi energy is comparable to the superconducting energy gap, the superconductivity in FeSe is believed to be situated near the BCS-BEC crossover [59]. Another advantage of FeSe is its non-magnetic properties under normal pressure, making it an ideal platform for investigating the nematic phase and superconductivity [60]. Additionally, SDW in high-quality FeSe single crystals under high-pressure was revealed for the first time [17], which was not observed in previous studies using impure phase samples [16]. Numerous studies have demonstrated that probing the intrinsic properties of FeSe heavily relies on the quality of the single crystals. Consequently, conducting a comprehensive review of FeSe single crystal growth is not only valuable in summarizing existing knowledge, but also offers significant guidance for future FeSe research endeavors.

### 2.1. Flux Method for Growing FeSe Single Crystals 

Maw-Kuen Wu’s group reported the observation of superconductivity with zero-resistance transition temperature at 8 K in the FeSe polycrystalline bulk for the first time [9]. The crystal of FeSe is composed of a stack of edge-sharing FeSe_4_-tetrahedra layer-by-layer, as shown schematically in Figure 1. An FeSe single crystal with a size about 500 μm was firstly synthesized using the flux method employing a NaCl/KCl mixed eutectic [61]. The preparation process can be divided into two stages. Firstly, Fe_1.2_Se polycrystalline with nominal stoichiometry was prepared through a traditional solid-state reaction using high purity Fe and Se powders as the raw materials. Then, the obtained Fe_1.2_Se polycrystal powder and NaCl/KCl mixed eutectic with mole ratio 1:1 were ground and sealed in an evacuated quartz tube. The quartz tube was slowly heated to 850 °C and kept two hours for sufficient solution of the raw materials and flux. Afterward, the temperature was gradually reduced at a rate of 3 °C/h down to 600 °C, followed by furnace cooling. FeSe single crystals were separated from the flux by dissolving the NaCl/KCl mixed eutectic in deionized water. 

Figure 2 illustrates the basic physical properties of the obtained single crystals. In Figure 2a, the optical image of FeSe reveals two different shapes present in all the grown single crystals: rectangular and hexagonal, both with a size of approximately 500 μm. The X-ray diffraction (XRD) pattern in Figure 2b shows two sets of peaks corresponding to two distinct crystal structures: tetragonal (with space groups P4/nmm) and hexagonal (with space groups P63/mmc). This indicates the presence of non-superconducting impurities in the single crystals. The temperature dependence of resistance and magnetic susceptibility is presented in Figure 2c,d, respectively. The large superconducting transition width (Δ*T*_c_) and the small superconducting volume fraction observed suggest a low-quality superconducting tetragonal phase.

Subsequently, several research groups successfully synthesized FeSe single crystals using similar methods [62,63,64,65,66]. While superconductivity has improved, the presence of impurities remains a significant concern. Impurities such as hexagonal FeSe, Fe_7_Se_8_ and Fe_3_O_4_ exist in all as-grown single crystals, posing a major obstacle in understanding the intrinsic properties of FeSe. The strong magnetism of these impurities often results in a prominent ferromagnetic background in the superconducting magnetization-field (*M*-*H*) loop [63]. FeSe single crystals with no impurity have been synthesized using the LiCl/CsCl flux method where the ferromagnetic background in *M*-*H* loop is nearly absent below the *T*_c_ [67]. However, even with this method, the superconducting volume fraction remains below 60%, indicating the need for further improvements.

### 2.2. Chemical Vapor Transport (CVT) Method for Growing FeSe Single Crystals 

Despite the successful growth of large-sized FeSe single crystals using the flux method, the quality of the crystals and the presence of impurities hindered related research. The iodine vapor transport method did not effectively improve the crystal quality [64]. However, a breakthrough was achieved through the preparation of high-quality FeSe single crystals using the CVT method with a KCl/AlCl_3_ transport agent [13,68]. A distinct kink at approximately 90 K was observed in the temperature dependence of resistance *R*(*T*) and was confirmed to be a structural(nematic) transition from the tetragonal to orthorhombic phase [12,68,69,70,71,72]. The temperature dependence of resistance and magnetic susceptibility confirmed the presence of a superconducting transition around 9.4 K. The superconducting transition width of about 1.5 K and a nearly 100% superconducting volume fraction demonstrated good superconductivity [68].

The synthesis process is as follows: High-purity Fe and Se powders were sealed in an evacuated quartz tube along with KCl and AlCl_3_ powders. The quartz tube was horizontally placed in a tube furnace with a double-temperature zone. The hot part of the tube containing the raw materials was heated to 390 °C while the cold part for single crystal growth was kept at 240 °C. After approximately 30 days of transport growth, a large number of single crystals with tetragonal morphology could be observed in the cold part. Similarly to the flux method, FeSe single crystals need to be separated from the flux by dissolving the KCl/AlCl_3_ mixed eutectic in deionized water. The schematic representation of the typical CVT growth assembly is shown in Figure 3a. The scanning electron microscope image in Figure 3b displays the clear layered structure of a tetragonal FeSe single crystal [68]. The temperature dependence of resistivity (*ρ*-*T*) and magnetization (*M*-*T*), shown in Figure 3c and inset, indicate high-quality crystallization and good bulk superconductivity [73].

Since then, the preparation of FeSe single crystals using similar methods has become more prevalent, leading to a flourishing research landscape, due to the availability of high-quality single crystals. However, the quality of these single crystals is highly sensitive to the preparation conditions, primarily influenced by the complex binary Fe-Se composition–temperature phase diagram [74]. 

A study conducted by A. E. Böhmer et al. explored the relationship between transition temperatures and residual resistivity ratio (RRR) in vapor-grown FeSe [75]. Their findings revealed that the inclusion of some excess Fe, with an Fe:Se ratio of 1.1:1 as nominal compositions, effectively suppressed the formation of the hexagonal Fe_7_Se_8_ phase. Additionally, the temperature conditions during growth strongly influenced the single crystal quality, with an optimal temperature gradient of 350–390 °C observed in their work. In addition, the tilt angle of the quartz tube can also have some impact on the growth. Figure 4a shows the single crystals under the optimal growth conditions and the schematic picture. Figure 4b,c provides a summary of the correlation between RRR (ratio of resistance at 250 K to resistance just above the *T*_c_), *T*_s_ and *T*_c_. Both the *T*_s_ and *T*_c_ decrease as the RRR increases. Composition analysis using wavelength dispersive X-ray spectroscopy (WDS) indicated no correlation between the *T*_c_ and sample composition. Extrapolating the linear relation between the *T*_s_ and *T*_c_ suggests that superconductivity would be completely suppressed when the *T*_s_ reaches 64 K.

## 3. Single Crystal Growth and Superconductivity of FeSe_1−*x*_S*_x_*

The nematicity in FeSe_1−*x*_S*_x_* is significantly suppressed with S doping, which completely disappears at *x* = 0.17. As the S content increases, the nematic fluctuations are strongly enhanced, and the nematic susceptibility diverges as it approaches *T* = 0, indicating the presence of a nematic QCP at *x* = 0.17 [76]. Notably, no AFM fluctuations are observed at the nematic QCP, suggesting a distinct separation between the nematicity and magnetic order. Consequently, the FeSe_1−*x*_S*_x_* system proves to be an excellent platform for studying the relationship between the nematicity and superconductivity. Furthermore, the non-Fermi liquid behavior at QCP indicates that nematic critical fluctuations have a significant influence on the normal-state electronic properties [32]. Magnetotransport behavior deviates significantly from the Fermi liquid and linear resistivity at low temperatures within the nematic phase suggest the presence of scattering from low-energy spin fluctuations [31,77,78]. These phenomena provide compelling evidence for the intrinsic connection between quantum criticality, strange metal state, and unconventional superconductivity in the FeSe_1−*x*_S*_x_* system.

### 3.1. CVT Growth of FeSe_1−x_S_x_ Single Crystals with Low S Doping

FeSe_1−*x*_S*_x_* single crystals are typically grown by CVT from FeSe up to *x* ~ 0.4, using similar preparation methods as FeSe [31,76,77,79,80]. Figure 5a displays the temperature dependence of the resistivity normalized to the value at 300 K value for *x* = 0 to 0.25. With S doping, a clear kink in resistivity gradually decreases to lower temperatures and disappears at a nonmagnetic nematic QCP, *x* = 0.17, as shown more clearly in Figure 5b, depicting the temperature dependence of the first derivative d*ρ*/d*T*. The discovery of QCP with nonmagnetic nematicity in the 11 system has raised the prospect of investigating the role of the relationship between nematicity and superconductivity [32,81].

### 3.2. Hydrothermal Method for Growing FeSe_1−x_S_x_ Single Crystals across the Entire Doping Range

While the CVT method allows the synthesis of FeSe_1−*x*_S*_x_* single crystals with *x* ≤ 0.29, the hydrothermal method has been employed to overcome this limitation. Xiaofang Lai et al. successfully synthesized tetragonal FeS through the hydrothermal reaction of Fe powder with a sulfide solution and observed bulk superconductivity for the first time at 5 K [34]. Subsequently, a hydrothermal ion release/introduction technique involving the de-intercalation of K ions from K_0.8_Fe_1.6_Se_2−*x*_S*_x_* precursors has been widely utilized for the preparation of FeS and FeSe_1−*x*_S*_x_* single crystals [33,82,83,84,85,86,87,88], as schematically depicted in Figure 6a. The process involves the growth of K_0.8_Fe_1.6_Se_2−*x*_S*_x_* precursors using the self-flux method, followed by the addition of Fe powder, selenourea, thiourea, and K_0.8_Fe_1.6_Se_2−*x*_S*_x_* single crystals pieces to a solution containing dissolved NaOH in deionized water within a Teflon-linked stainless-steel autoclave (25 mL). The autoclave is then sealed and heated to 130–150 °C for 50–70 h resulting in the formation of FeSe_1−*x*_S*_x_* single crystals, as shown in Figure 6b. 

Figure 7 presents a comprehensive phase diagram of FeSe_1−*x*_S*_x_* single crystals, encompassing the entire region obtained from the hydrothermal method [33] and a partial region (0 ≤ *x* ≤ 0.29) obtained from the CVT method [31,76,77]. The values of the *T*_s_ and *T*_c_ obtained from the hydrothermal method are slightly lower than those from the CVT method, possibly due to disorder effects in the crystals [75]. The exponent “*n*” in the contour plot corresponds to the power law, *ρ* (*T*) = *ρ*_0_ + *AT^n^*, where *ρ*_0_ represents the residual resistivity. In the nematic phase, the resistivity exhibits a non-Fermi liquid behavior characterized by sublinear temperature dependence. Outside the nematic phase, the resistivity at low temperatures follows a prefect Fermi liquid behavior, i.e., *T*^2^ dependence. In the Fermi liquid region, the coefficient *A* decreases monotonically with S doping, indicating a reduction in effective mass, since *A* is proportional to the carrier effective mass according to the Landau Fermi liquid theory. Below the characteristic temperature *T*^*^, the resistivity displays an anomalous upturn just before the superconducting transition. The origin of this anomaly may be attributed to local magnetic impurity scattering or inelastic scattering due to crystallographic disorder.

## 4. Single Crystal Growth and Superconductivity of FeSe_1−*x*_Te*_x_*

Similar to S doping, the nematicity in FeSe_1−*x*_Te*_x_* is gradually suppressed with Te doping and disappears at *x* = 0.5 [36,37]. The presence of nematic QCP accompanied by the superconducting dome is supported by the behavior of the nematic susceptibility in FeSe_1−*x*_Te*_x_* single crystals [89]. The magnetic order disappears under high pressure when *x* > 0.1, while the superconducting dome persists, suggesting that the enhancement of superconductivity in FeSe_1−*x*_Te*_x_* is not attributed to magnetism but rather to the nematic fluctuations [37]. In the case of higher Te content, FeSe_1−*x*_Te*_x_* exhibits topological surface superconductivity and the presence of Majorana fermions, making it the first high-temperature topological superconductor to be discovered [90,91]. In the region near FeTe, a competition between magnetism and superconductivity is also observed [40,42]. The magnetism in FeSe_1−*x*_Te*_x_* exhibits a bi-collinear antiferromagnetism, which is distinct from the collinear antiferromagnetism observed in iron-pnictides [43]. Additionally, FeSe_1−*x*_Te*_x_* displays an excellent high upper critical field and low anisotropy, which significantly reduce the challenges associated with applications [92]. Researchers have successfully overcome the effects of excess iron and, more recently, phase separation, and the intrinsic properties of FeSe_1−*x*_Te*_x_* are gradually being unveiled.

### 4.1. CVT Growth of FeSe_1−x_Te_x_ (0 ≤ x ≤ 0.5) Single Crystals

While high-quality single crystals of FeSe_1−*x*_S*_x_* have been successfully obtained, achieving homogenous Te-doping single crystals remains challenging due to strict preparation conditions and the phase separation in the region of 0.1 ≤ *x* ≤ 0.4 [38,51,52]. In recent years, significant efforts have been made in crystal growth, leading to several studies on phase separation regions. The synthesis of FeSe_1−*x*_Te*_x_* (0 ≤ *x* ≤ 0.41) single crystals using the flux method with a temperature gradient, including the phase separation regions, has been reported for the first time [36]. 

Figure 8a illustrates the schematic diagram of the growth setup, where a horizontal quartz tube is placed in a two-temperature zone tube furnace. The mixture of high-purity Fe, Se and Te powders, pre-sintered at 450 °C, along with a flux mixture of AlCl_3_/KCl was placed in high-temperature zone of quartz tube. After 20–30 days, flake-like single crystals were obtained in the low-temperature zone and the residual flux was removed by dissolving it in distilled water, as shown in Figure 8b. Then, a FeSe_0.67_Te_0.33_ single crystal was grown using a flux method with a single-temperature zone in a box furnace [38]. 

The results of these two works are summarized in a phase diagram, shown in Figure 8c. The *T*_c_ exhibits a minimum around *x* ~ 0.2, which is attributed to the effect of sample disorder, as indicated by the relatively small RRR value [36,75]. The *T*_s_ decreases linearly with increasing Te doping and disappears at approximately *x* ~ 0.5. The *T*_c_ exhibits a maximum around *x* ~ 0.6, and the Néel temperature (*T*_N_) starts to appear when *x* > 0.9, accompanied by the suppression of superconductivity [93]. The breakthrough in the phase separation region provides a promising approach for the preparation of high-quality single crystals, particularly in the phase separation region, enabling the investigation of the evolution of the intrinsic properties of FeSe_1−*x*_Te*_x_* with Te doping.

Recently, significant progress has been made in the growth of high-quality FeSe_1−*x*_Te*_x_* (0 ≤ *x* ≤ 0.5) single crystals using the CVT method, and the temperature–composition phase diagrams have been established, as shown in Figure 9 [37]. Similar to the flux method with a two-temperature zone described earlier, the mixture of Fe, Se, and Te powders was sealed in a quartz ampoule with transport agents AlCl_3_/KCl and the growth time was 1–2 weeks. The temperatures of the hot and cold sides were controlled at 420 and 250 °C for 0 ≤ *x* ≤ 0.25 (620 and 450 °C for 0.25 ≤ *x* ≤ 0.55), respectively, which play a significant role in the crystal growth process.

Despite the similar synthesis methods employed by different research groups, there is considerable variation in the quality of the obtained single crystals, including RRR, the superconducting transition temperature *T*_c_ and transition width Δ*T*. In this systematic study, a comprehensive analysis of RRR with a large number of data points, represented by *ρ*(200 K)/*ρ*(15 K), reveals a monotonous decrease with increasing Te concentration, as shown in Figure 9c. This suggests an intrinsic origin of the minimum *T*_c_ observed at *x* = 0.3. Additionally, when considering the temperature–pressure–composition phase diagrams of FeSe_1−*x*_Te*_x_* (0 ≤ *x* ≤ 0.5) single crystals, it is proposed that nematic fluctuations play a role in enhancing the *T*_c_ above *x* = 0.3 and contribute to the formation of the observed *T*_c_-dip. 

### 4.2. Self-Flux Plus Annealing Method for Growing FeSe_1−x_Te_x_ (0.5 < x ≤ 1) Single Crystals

FeSe_1−*x*_Te*_x_* (0.5 < *x* ≤ 1) single crystals can be grown using standard melting methods, such as the Bridgeman method [50], self-flux method (a modified Bridgeman method, similar to each other) [94,95] and optical zone melting [96]. In the self-flux method, high-purity Fe, Se and Te powders with nominal ratios were loaded into a quartz tube, which wa then evacuated and sealed. To prevent cracking during the growth process, it is necessary to seal the quartz tube into a lager quartz tube. The assembly was slowly heated to 1050 °C and sustained for 24 h, followed by cooling down to 710 °C at a rate of 3 °C/h and furnace cooling. The obtained single crystals have a mirror-like surface and can reach the centimeter scale sizes, as shown in Figure 10a [97].

The position of excess iron in the crystal structure is shown in Figure 10b, marked by the orange ball. Excess Fe in the crystal structure of FeSe_1−*x*_Te*_x_* significantly affects its intrinsic properties, such as localization of the charge carriers [53,54,55], spin glass phase [56] and incoherent electronic states [54,57]. Annealing processes have been developed to effectively remove excess Fe. FeTe_0.61_Se_0.39_ single crystals were successfully annealed in a vacuum environment for the first time at 400 °C for more than 10 days, resulting a sharp superconducting transition at around 14 K [99]. Subsequently, vacuum annealing techniques have been applied to remove excess Fe from FeSe_1−*x*_Te*_x_* (0.5 < *x* ≤ 1) single crystals [93,100]. It was reported that N_2_ annealing can also effectively remove excess Fe [101]. However, it was later discovered that vacuum and N_2_ annealing have no effect on the excess Fe, and the observed improvement was actually due to the action of a small amount of residual O_2_ present during the annealing process [102]. Apart from O_2_ annealing, elements such as Te, Se, S, P, As, I, and Sb have been proven to effectively remove excess Fe through vapor annealing for FeSe_1−*x*_Te*_x_* (0.5 < *x* ≤ 1) single crystals [103,104,105,106,107,108]. For efficiency and nontoxicity, we focus on providing a detailed introduction using O_2_ annealing to remove excess Fe.

Figure 11 shows the schematic picture of the annealing system used for O_2_ [41]. To perform the O_2_ annealing, as-grown single crystals were cut and cleaved into thin slices with dimensions of about 2.0 × 1.0 × 0.05 mm^3^. These slices were then weighed and loaded into a quartz tube with an inner diameter of 10 mm. The quartz tube was carefully evacuated using a diffusion pump, and the pressure in the tube was detected using a diaphragm-type manometer with an accuracy greater than 1 mTorr. Once the gas was fully removed, the quartz tube was filled with Ar/O_2_ (1% Ar) mixed gas and sealed to a length of 100 mm. The pressure in the system is continuously monitored during the sealing process to prevent gas leakage and control the O_2_ pressure in the quartz tube. The crystals were then annealed at 400 °C for various periods of time and subsequently quenched in water.

The doping–temperature phase diagram for the as-grown and annealed Fe_1+*y*_Te_1−*x*_Se*_x_* (0 ≤ *x* ≤ 0.43, *y* represents excess Fe) were established based on the magnetization, magnetic susceptibility, resistivity, and Hall effects, as shown in Figure 12a,b, respectively [39]. In the as-grown, there is a clear spin glass state originating from excess Fe in the interstitial site before the onset of superconductivity. The superconductivity observed in the as-grown crystals is not of bulk nature and can only be obviously detected through the temperature dependence of resistivity. After annealing, significant changes in superconductivity and magnetic order are observed. The AFM phase is suppressed into a very narrow regions for *x* (Se) < 0.05, and the spin glass state completely disappears. This confirms the effective removal of excess Fe through annealing. The superconducting state exhibits a clear bulk effect and can be easily detected by magnetic measurements. 

In our recent work, we have successfully prepared high-quality full-range FeSe_1−*x*_Te*_x_* single crystals, with varying Te doping levels (0 ≤ *x* ≤ 0.5 by CVT and 0.5 < *x* ≤ 1 by the flux method plus annealing). The corresponding phase diagram is illustrated in Figure 13. Notably, Te doping gradually suppresses the nematic phase until it completely disappears at *x* = 0.5. Our results also reveal that the *T*_c_ reaches its minimum at *x* = 0.3, which aligns with the findings of Mukasa et al. [37], further supporting the intrinsic nature of the *T*_c_-dip phenomenon observed in FeSe_1−*x*_Te*_x_*. Subsequently, the *T*_c_ increases and reaches a maximum at *x* = 0.6 but gradually decreases upon further Te doping, eventually leading to a transition into a non-superconducting antiferromagnetic state.

The high chemical stability, high *T*_c_, and strong upper critical field exhibited by FeSe_1−*x*_Te*_x_* single crystals make them excellent candidates for investigating the pairing mechanism underlying high-temperature superconductivity. Consequently, the comprehensive phase diagram we have established for FeSe_1−*x*_Te*_x_* provides valuable support for the ongoing exploration of the superconducting pairing mechanism in high-temperature superconductors.

### 4.3. Optical Zone-Melting Technique for Growing FeSe_1−x_Te_x_ Single Crystals

FeSe_1−*x*_Te*_x_* single crystals also can be grown using the optical zone-melting technique [96,109]. This method allows for real-time observation of single crystal growth and precise control of the growth rate by visualizing the melting zone. Figure 14 illustrates the schematic picture of a single crystal growth and shows a large-sized single crystal obtained using this technique. The growth process is as follows:

High-purity powders of Fe, Se and Te with a nominal ratio were mixed in a ball mill for 4 h. The mixed powders were cold pressed into discs under a uniaxial pressure of 400 kg·cm^−2^, and then heated at 600 °C for 20 h under a vacuum. The reacted bulk material was reground into a fine powder and loaded into a double quartz tube. The tube was loaded in an optical zone-melting furnace equipped with two 1500 W halogen lamps as infrared radiation sources, as shown in Figure 14. The tube was rotated at a rate of 20 rpm and moved at a rate of 1–2 mm·h^−1^. After the growth, the as-grown crystals undergo an annealing process: ramping to 700–800 °C in 7 h, holding for 48 h; cool to 420 °C in 4 h, hold for 30 h; and finally shutting down the furnace and cooling to room temperature.

Despite obtaining large-sized and well-crystallized single crystals using the optical zone-melting technique, the upwarping behavior of the *R*(*T*) curves before superconducting transition is still apparent, indicating the presence of excess Fe in the crystals [96]. Moreover, due to the complexity of the preparation process and the more established self-flux method, the optical zone-melting method is not commonly used for the growth of FeSe_1−*x*_Te*_x_* single crystals. 

## 5. Single Crystal Growth and Superconductivity of FeTe_1−*x*_S*_x_*

FeTe_1−*x*_S*_x_* system also exhibits superconductivity. Yoshikazu Mizuguchi et al. first reported the superconductivity in the FeTe_1−*x*_S*_x_* system and found that the *T*_c_ can reach 10 K when *x* is 0.2 [110]. FeTe_1−*x*_S*_x_* single crystals with low S doping were grown using the self-flux method, similar to FeSe_1−*x*_Te*_x_* (0.5 < *x* ≤ 1) single crystals [111,112,113,114,115]. Annealing treatment is also necessary to improve superconductivity for FeTe_1−*x*_S*_x_* single crystals, although the excess Fe cannot be completely removed [116,117,118,119,120,121]. The solubility limit of S in FeTe is about 12% and Chiheng Dong et al. provided the phase diagram in this region [119,122]. With S doping, AFM is suppressed and superconductivity is enhanced. 

Caiye Zhao et al. successfully synthesized a series of FeS_1−*x*_Te*_x_* (0 ≤ *x* ≤ 0.15) single crystals by a hydrothermal method for the first time and provided a phase diagram of FeS_1−*x*_Te*_x_* single crystals, shown in Figure 15 [123]. The *T*_c_ is rapidly suppressed with the Te doping for FeS_1−*x*_Te*_x_* (0 ≤ *x* ≤ 0.15) single crystals and finally disappears when *x* > 0.1. Due to the large solution limited region, only a small amount of doping can be applied at both ends of the phase diagram. The complete phase diagram needs further exploration. 

## 6. Conclusions

In conclusion, significant progress has been made in the preparation of 11 system single crystals, including FeSe_1−*x*_Te*_x_* and FeSe_1−*x*_S*_x_*, through various methods. A comprehensive phase diagram has been constructed, as depicted in Figure 16, summarizing the superconducting transition temperatures (*T*_c_), the onset of nematic phase (*T*_s_), and the Néel temperature (*T*_N_) for the single crystals prepared using the optimal techniques in different intervals. 

High quality FeSe_1−*x*_S*_x_* (0 ≤ *x* ≤ 0.29) and FeSe_1−*x*_Te*_x_* (0 ≤ *x* ≤ 0.55) single crystals are typically grown using CVT method with AlCl_3_/KCl transport agent. It is fortuitous that the range encompassing these single crystals includes the nematic phase without magnetic order. The exceptional quality of these crystals serves as an excellent platform for investigating the interplay between nematicity and superconductivity. FeSe_1−*x*_S*_x_* (0.29 ≤ *x* ≤ 1) single crystals, however, can only be synthesized using hydrothermal method. Although the quality of single crystals using hydrothermal is slightly inferior to those grown using CVT, they still hold great significance for studying the complete phase diagram of FeSe_1−*x*_S*_x_*. By utilizing the self-flux plus annealing technique, single crystals without excess Fe in the highly Te doping region can be obtained. In this particular region, the *T*_c_ reaches maximum of the entire phase diagram, approximately 15 K, occurring around *x* (Te) ~ 0.6. Furthermore, AFM state is observed within a narrow region around FeTe.

The connection between the ordered states and superconductivity have not been well resolved, and the relationship between nematicity and SDW has been described as a “chicken-egg” problem [10]. Understanding the interplay between these states is complex and challenging. Furthermore, the behavior of superconductivity throughout the entire phase diagram presents intricate twists and turns, adding to the puzzle. In summary, the establishments of the comprehensive phase diagram for the 11 iron-based system is of utmost importance for unraveling the mechanism behind high-temperature superconductivity and for discovering novel superconducting materials. 

## Figures and Tables

**Figure 1 materials-16-04895-f001:**
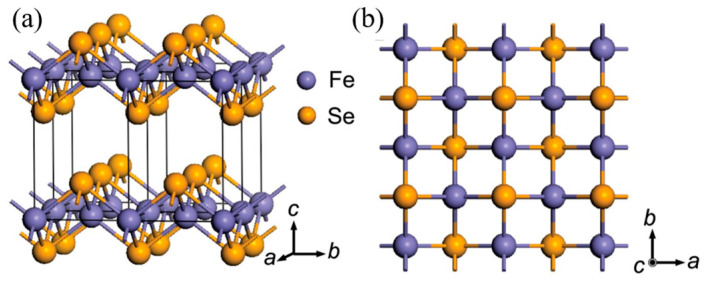
Crystal structure of tetragonal FeSe. (**a**) Perspective view along the *a* axis. (**b**) Parallel view along the *c* axis. Reprinted with permission from Ref. [9]. Copyright 2008, copyright the National Academy of Sciences of the USA.

**Figure 2 materials-16-04895-f002:**
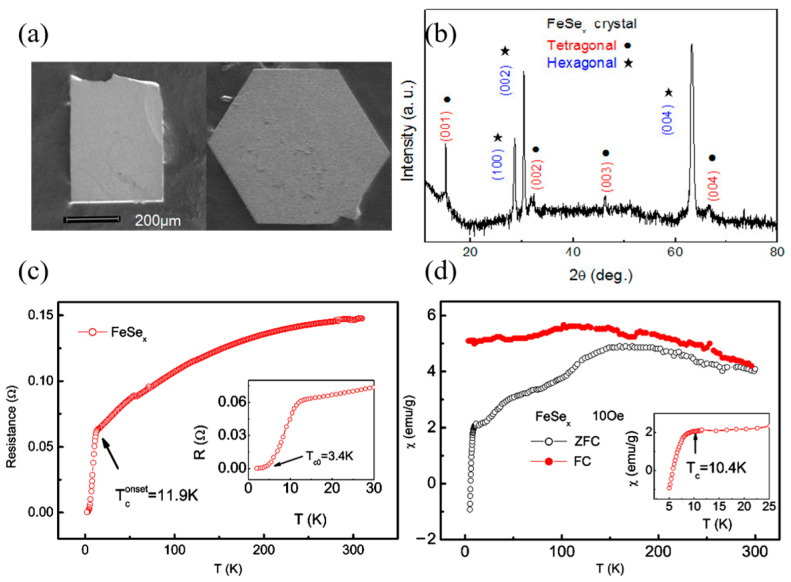
(**a**) Optical image of FeSe with two different shapes, rectangular and hexagonal; (**b**) XRD pattern from FeSe flake along *c* axis, including two sets of peaks; (**c**) Temperature dependence of resistance for FeSe single crystal in the *ab* plane, the inset is a magnified plot in the low temperature region; (**d**) Temperature dependence of magnetic susceptibility for FeSe single crystal at 10 Oe, the inset is a magnified plot in the low temperature region. Reprinted with permission from Ref. [61]. Copyright 2008, copyright IOP Publishing Ltd.

**Figure 3 materials-16-04895-f003:**
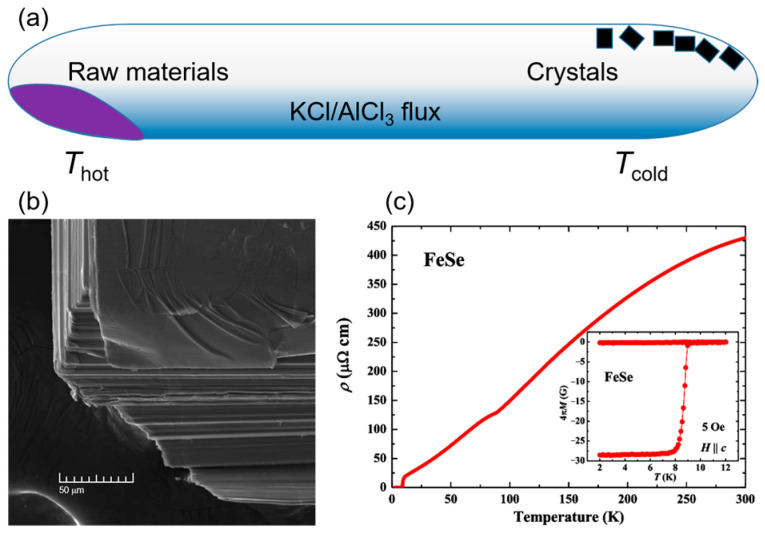
(**a**) Schematic image of the typical CVT growth assembly. (**b**) The scanning electron microscope image of the layered structure of a tetragonal FeSe single crystal. Reprinted with permission from Ref. [68]. Copyright 2013, copyright the Royal Society of Chemistry. (**c**) Temperature dependence of resistivity for FeSe single crystal. The inset shows the magnetic susceptibility measured under *H* = 5 Oe external magnetic field. Reprinted with permission from Ref. [73]. Copyright 2015, copyright the American Physical Society.

**Figure 4 materials-16-04895-f004:**
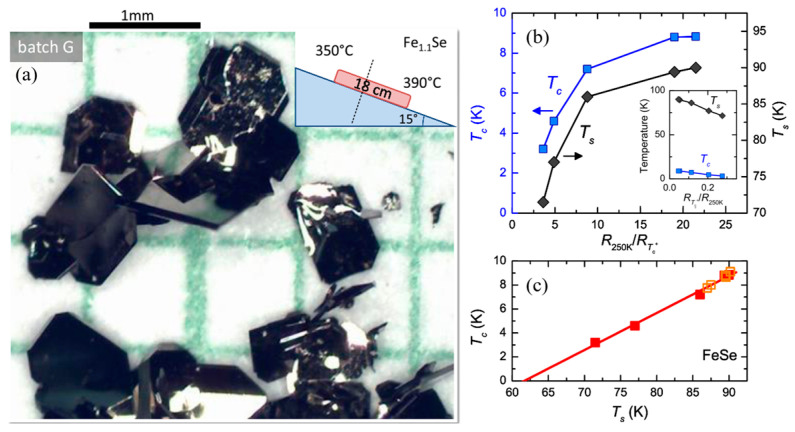
(**a**) Photograph of tetragonal FeSe single crystals under optimal growth conditions and the schematic picture. (**b**) Structural transition temperature *T*_s_ and superconducting transition temperature *T*_c_ as a function of residual resistivity ratio (ratio of resistance at 250 K to resistance just above *T*_c_) for different samples. The inset shows the transition temperature as a function of the inverse residual resistivity ratio. (**c**) *T*_c_ as a function of *T*_s_ for various samples, Red squares show data from panel (**b**), and Orange squares represent data on samples grown as part of earlier studies in Ref. [13]. Reprinted with permission from Ref. [75]. Copyright 2016, copyright the American Physical Society.

**Figure 5 materials-16-04895-f005:**
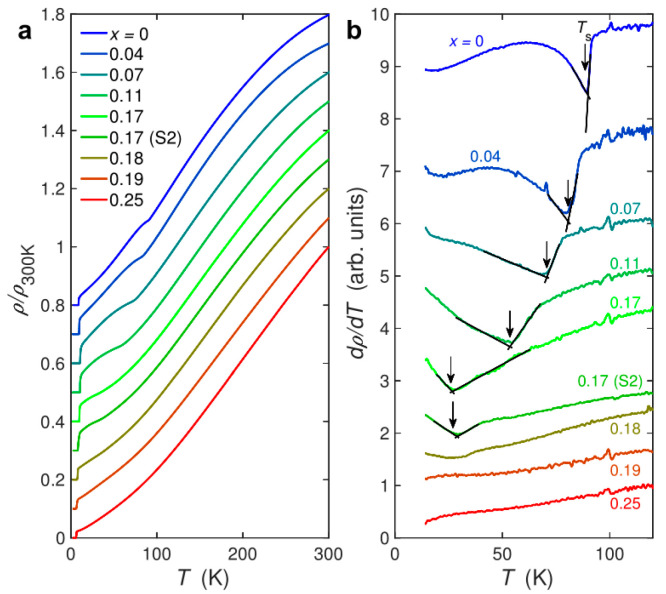
Temperature dependence of the resistivity of FeSe_1−*x*_S*_x_*. (**a**) Temperature dependence of resistivity normalized to the 300 K value from *x* = 0 to 0.25. (**b**) The first derivative of the resistivity with respect to temperature for the same data. The curves for different S concentrations have been offset for clarity. The location of the structural transition, *T*_s_, is defined by the intercept of the linear fits on either side of the transition, as indicated by arrows [77].

**Figure 6 materials-16-04895-f006:**
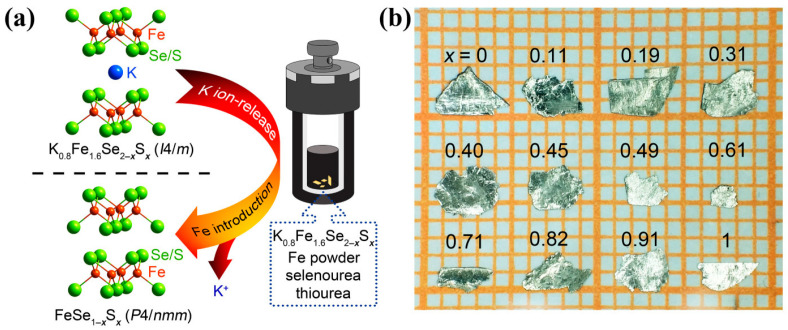
(**a**) Schematic illustration of the hydrothermal ion release/introduction route for the synthesis of FeSe_1−*x*_S*_x_* single crystals. (**b**) Optical image of select FeSe_1−*x*_S*_x_* single crystals. Reprinted with permission from Ref. [33]. Copyright 2021, copyright the American Physical Society.

**Figure 7 materials-16-04895-f007:**
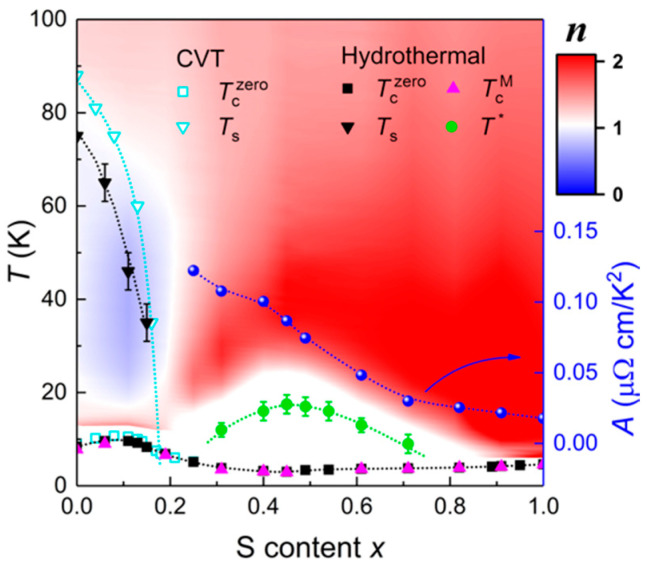
Complete phase diagram of FeSe_1−*x*_S*_x_* single crystals. *T*_s_ represents the nematic transition temperature. $T_{c}^{\mathrm{zero}}$ and $T_{c}^{M}$ are the SC transition temperatures obtained from resistivity and magnetization measurements, respectively. *T** is the characteristic temperature at which the *ρ*-*T* curves show local minima at low temperatures. Reprinted with permission from Ref. [33]. Copyright 2021, copyright the American Physical Society.

**Figure 8 materials-16-04895-f008:**
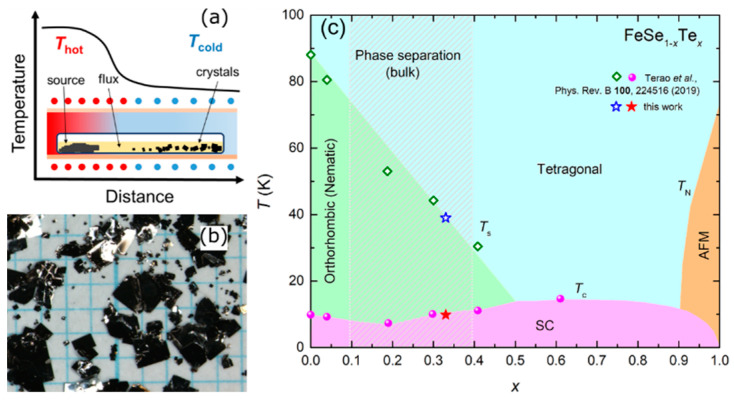
(**a**) Schematic image of the temperature distribution in the horizontal tube furnace for single-crystal growth of FeSe_1−*x*_Te*_x_* by the flux method. Reprinted with permission from Ref. [36]. Copyright 2019, copyright the American Physical Society. (**b**) Photograph of as-grown single crystals of FeSe_1−*x*_Te*_x_* after removing the flux. Reprinted with permission from Ref. [36]. Copyright 2019, copyright the American Physical Society. (**c**) Complete temperature-doping *x* phase diagram of FeSe_1−*x*_Te*_x_* single crystals. Reprinted with permission from Ref. [38]. Copyright 2021, copyright the IOP Publishing.

**Figure 9 materials-16-04895-f009:**
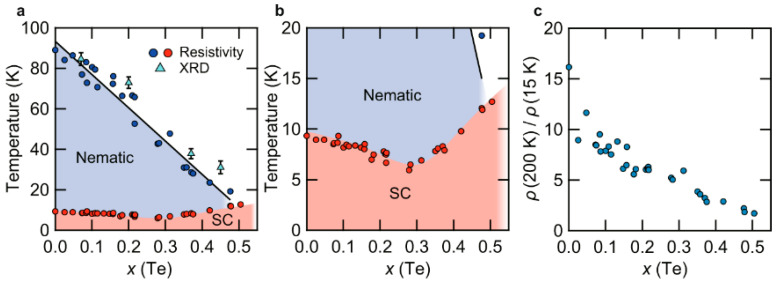
(**a**) Temperature—*x* (Te) phase diagram of FeSe_1−*x*_Te*_x_* (0 ≤ *x* ≤ 0.5) single crystals. (**b**) The same as in (**a**), but the temperature range is 0–20 K. (**c**) Dependence of *ρ*(200 K)/*ρ*(15 K) on *x* (Te) extracted from the resistivity data [37].

**Figure 10 materials-16-04895-f010:**
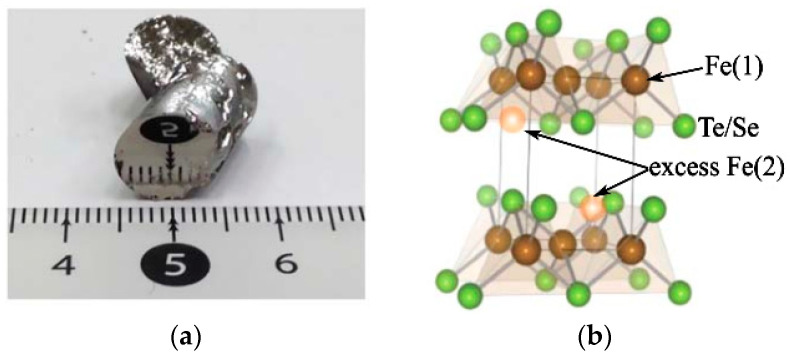
(**a**) Photograph of the as-grown FeTe_0.6_Se_0.4_ single crystal. Reprinted with permission from Ref. [97]. Copyright 2015, copyright the IOP Publishing, Ltd. (**b**) Crystal structure of FeSe_1−*x*_Te*_x_*. The green ball represents the Fe(1) in tetragonal lattice, and the orange ball represents the excess Fe(2) [98]. Reprinted with permission from Ref. [41]. Copyright 2019, copyright the IOP Publishing, Ltd.

**Figure 11 materials-16-04895-f011:**
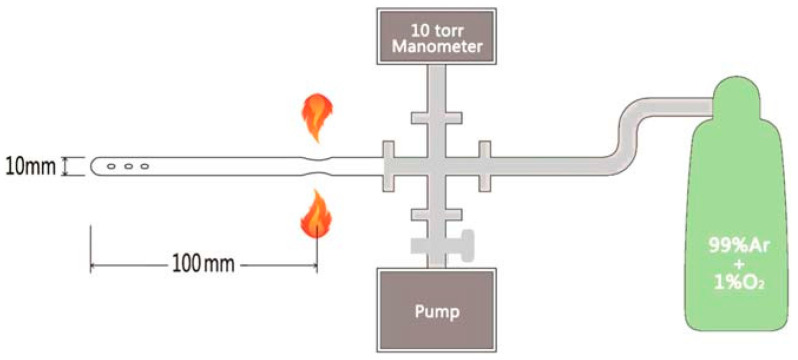
Schematic picture of the annealing system for sealing the crystal in quartz tube with a controlled amount of O_2_. Reprinted with permission from Ref. [41]. Copyright 2019, copyright the IOP Publishing, Ltd.

**Figure 12 materials-16-04895-f012:**
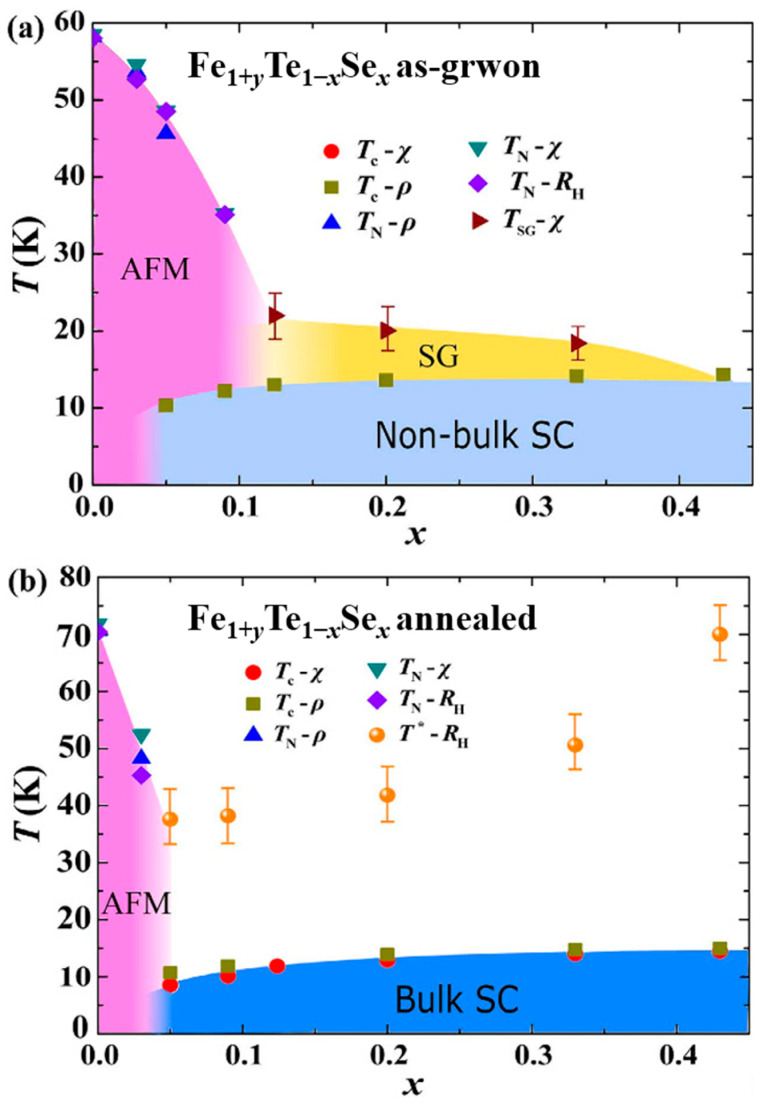
The doping–temperature (*x*-*T*) phase diagrams for Fe_1+*y*_Te_1−*x*_Se*_x_* (0 ≤ *x* ≤ 0.43, *y* represents excess Fe) single crystals (**a**) before and (**b**) after O_2_ annealing obtained from magnetization, magnetic susceptibility, resistivity and Hall effect measurements [39].

**Figure 13 materials-16-04895-f013:**
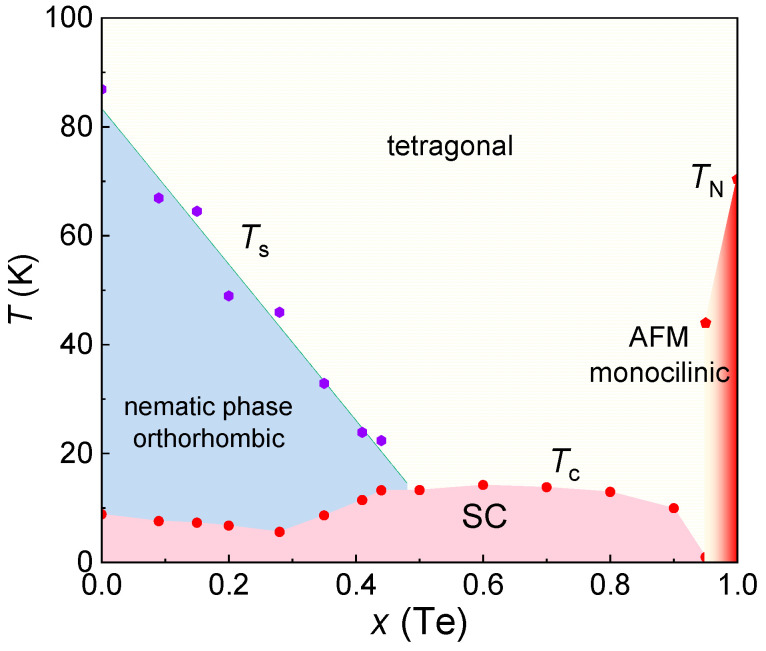
Complete phase diagram of FeSe_1−*x*_Te*_x_* (0 ≤ *x* ≤ 1) single crystals in our recent work.

**Figure 14 materials-16-04895-f014:**
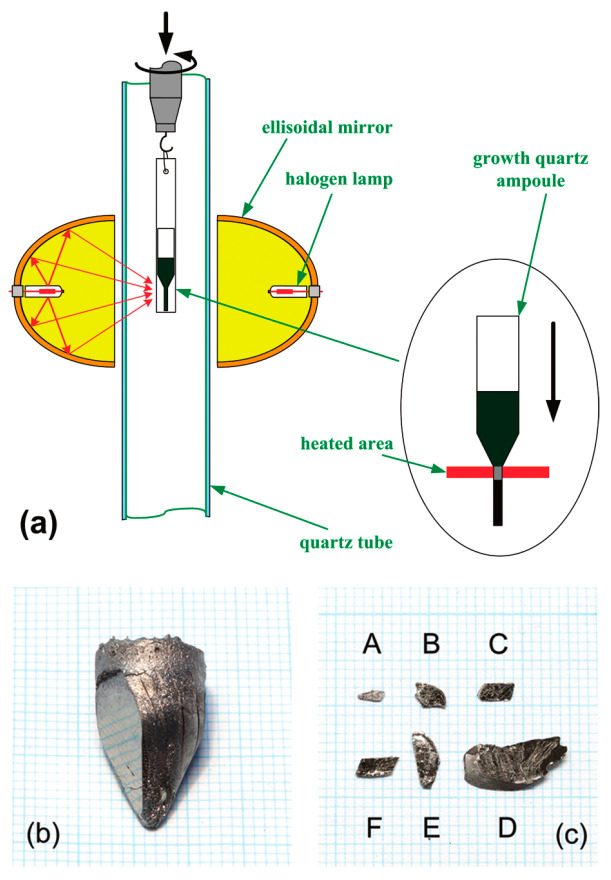
(**a**) Schematic diagram of apparatus setup of the optical zone-melting method. The red arrows represent the light path of the light source, and the black arrow represents the direction of single crystal growth. (**b**) Single crystal boule of as-grown FeTe_0.7_Se_0.3_ single crystal on a 1 mm grid. The shiny surface is the a–b plane. (**c**) The crystal flakes with the (001) face. Crystals from A–F represent FeSe*_x_*Te_1−*x*_ single crystals of *x* = 0.3, 0.5, 0.6, 0.7, 0.9, and 1.0, respectively. Reprinted with permission from Ref. [96]. Copyright 2009, copyright the American Chemical Society.

**Figure 15 materials-16-04895-f015:**
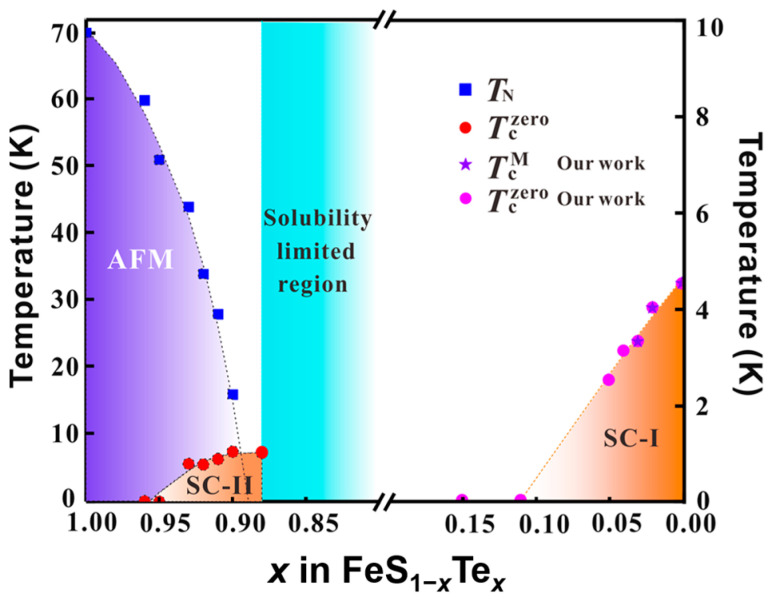
The doping phase diagram of FeS_1−*x*_Te*_x_* single crystals [119,123].

**Figure 16 materials-16-04895-f016:**
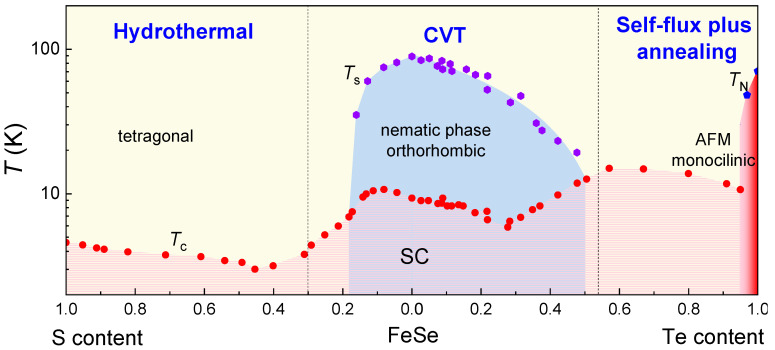
The entire phase diagram of FeSe_1−*x*_Te*_x_* and FeSe_1−*x*_S*_x_* single crystals synthesized by the optimal methods, hydrothermal for FeSe_1−*x*_S*_x_* (0.29 ≤ *x* ≤ 1) [33], CVT for FeSe_1−*x*_S*_x_* (0 ≤ *x* ≤ 0.29) [31,76,77] and FeSe_1−*x*_Te*_x_* (0 ≤ *x* ≤ 0.55) [37] and self-flux plus annealing for FeSe_1−*x*_Te*_x_* (0.55 ≤ *x* ≤ 1) [39].

## Data Availability

The data used to support the findings of this study are available upon request from the corresponding author.

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
