# Peer review of "Review of Single Crystal Synthesis of 11 Iron-Based Superconductors"

_materials, 2023, doi:10.3390/ma16144895_

Round 1

Reviewer 2 Report

See Review document attached

No comment

Reviewer 3 Report

The manuscript titled "Review of single crystal synthesis of 11 Fe-based superconductors" provides a nice, comprehensive review of the preparation of single crystals in the 11 system, including FeSe1-xTex and FeSe1-xSx, using various methods.  The text is well-written in readable English and has a good structure, following a chronological order. I particularly liked Figure 16. The images are taken directly from the original articles without any alterations. I'm unsure whether permission from the authors of the original articles is required to publish them. However, I do have a few suggestions for typo corrections (listed below), and overall, I recommend publishing the manuscript in MDPI Materials.

  1. Line 28: “trtrahedral” Â’ “tetrahedral”
  2. Line 127: The period after the sentence is missing.
  3. Line 245: Missing space between “Figure” and “8”
  4. Why are there dots before the references in almost all images? I don't understand their meaning. Please remove them in the following way:

Change: "Figure 1. Crystal structure of tetragonal FeSe. [1]."

To: "Figure 1. Crystal structure of tetragonal FeSe [1]."

Author Response

I am sincerely grateful to the reviewer for recognizing and appreciating our work, and I feel deeply honored.

We have obtained all the necessary permissions to include the figures directly taken from the original articles in our manuscript. Additionally, we have carefully addressed the reviewer's suggestions regarding spelling mistakes, making sure to correct them one by one in the latest version of the manuscript. We truly appreciate the reviewers' meticulous attention to detail and their valuable corrections.

Once again, I extend my heartfelt thanks to the reviewers for recognition, appreciation, and helpful suggestions.

Round 2

Reviewer 1 Report

I am satisfied by the changes the authors made in the revised version

Reviewer 2 Report

Revisions have improved the manuscript. The review may be useful to researchers working on iron-based superconductors. The paper can be published as is.